# Dialogue Medical Information Extraction with Medical-Item Graph and Dialogue-Status Enriched Representation

**Lei Gao,[*] Xinnan Zhang,[*] Xian Wu,[†] Shen Ge and Yefeng Zheng**
Tencent Youtu Lab, Jarvis Research Center
{lexlgao, xinnanzhang, kevinxwu, shenge, yefengzheng}@tencent.com

## Abstract

The multi-turn doctor-patient dialogue includes rich medical knowledge, like the symptoms of the patient, the diagnosis and medication suggested by the doctor. If mined and represented properly, such medical knowledge can benefit a large range of clinical applications, including diagnosis assistance and medication recommendation. To derive structured knowledge from free text dialogues, we target a critical task: the Dialogue Medical Information Extraction (DMIE). DMIE aims to detect pre-defined clinical meaningful medical items (symptoms, surgery, etc.) as well as their statuses (positive, negative, etc.) from the dialogue. Existing approaches mainly formulate DMIE as a multi-label classification problem and ignore the relationships among medical items and statuses. Different from previous approaches, we propose a heterogeneous graph to model the relationship between items. We further propose two consecutive attention based modules to enrich the item representation with the dialogue and status. In this manner, we are able to model the relationships among medical items and statuses in the DMIE task. Experimental results on the public benchmark data set show that the proposed model outperforms previous works and achieves the state-of-the-art performance.

## 1 Introduction

The digitization of medical systems over the past decade has accumulated massive medical data. Among various medical data, the multi-turn doctor-patient dialogue contains rich medical knowledge, like the suggested medical test according to patient symptom descriptions, the recommended medication according to doctor's diagnosis results, etc. If such medical knowledge can be properly extracted and represented, it can enable the development of a large range of clinical applications, like disease

| Doctor-Patient Conversation |
|---|
| Patient: 医生你好，频发室早是什么原因，严重吗？
(Hello, doctor. What is the reason for the frequent ventricular premature beats? Is it serious?) |
| Doctor: 我将为您提供就诊指导。频发室早是心脏传导系统出了问题。严重不严重要根据具体情况才能定。做了心电图了吗？
(I will provide you with online guidance for medical treatment. Frequent ventricular premature beats are problems with the conduction system of the heart. Whether it is serious or not can be determined according to the specific situation. Have you got an electrocardiogram?) |
| Patient: 血压低就会有危险吗？还没做心电图。治疗是怎样治疗的？
(Is there any danger from low blood pressure? I haven't done an electrocardiogram. How is it treated?) |
| Doctor: 血压偏低不一定有危险。最好做一下24小时动态心电图，了解一下全天室早早搏有多少，需要的话可以射频消融术治疗。
(Low blood pressure is not necessarily dangerous. It is better to do a 24-hour dynamic electrocardiogram to find out how many premature ventricular beats throughout the day. Radiofrequency ablation can be used for treatment if necessary.) |
| Patient: 好的，谢谢医生。
(OK, thank you doctor.) |

| | Extracted Labels | | |
|---|---|---|---|
| | Category | Item | Status |
| Label 1 | Symptom | Frequent Ventricular Premature Beats | Patient-Pos |
| Label 2 | Symptom | Hypotension | Unknown |
| Label 3 | Surgery | Radiofrequency Ablation | Doctor-Pos |
| Label 4 | Test | Electrocardiogram | Patient-Neg |
| Label 5 | Test | Electrocardiogram | Doctor-Pos |

Figure 1: An example of the DMIE task, translated from Chinese. Given the whole dialogue, five labels are extracted. Each label consists of a category, an item and its status. The label and the corresponding evidence in dialogue are highlighted with the same color.

diagnosis assistance and medication recommendation. To this end, we target a critical task: the Dialogue Medical Information Extraction (DMIE).

Given a multi-turn doctor-patient dialogue, the DMIE task aims to label this dialogue with pre-defined medical items and their corresponding statuses. We use an example from the MIEACL data set[1] (Zhang et al., 2020) to illustrate the DMIE task. As shown in Figure 1, this dialogue receives five labels in total. Each label consists of three sub-components: *Category*, *Item* and *Status*, and

---

[*]Equal contribution.
[†]Corresponding author.

[1]https://github.com/nlpir2020/MIE-ACL-2020

their combination provides detailed medical descriptions. As shown in Table 1, all items are grouped into four types of categories: *Symptom*, *Surgery*, *Test* and *Other info*. And the *Other info* include mental state, sleep situation, whether drinking and etc. In addition to items and their categories, the MIEACL dataset also defined five statuses to provide a fine-grained state for each item. For example, for an item whose category is *Symptom*, the status *Patient-Positive* indicates that this patient mentions he/she has this symptom, and the status *Doctor-Negative* indicates that the doctor denies this symptom in the dialogue. For an item with a category of *Test*, the status *Doctor-Positive* indicates that the doctor recommends the patient to take this test. For example, in Figure 1, Label 4 (*Test-Electrocardiogram:Patient-Neg*) indicates that the patient mentioned he/she did not have an electrocardiogram test before. It is worth mentioning that the statuses of an item in DMIE tasks are not mutually exclusive, like Label 4 and Label 5, an item can have two statuses at the same time.

Most existing approaches either focus on recognizing a subset of items or directly formulate DMIE as a classification task. Among existing DMIE works, Lin et al. (Lin et al., 2019) and Du et al. (Du et al., 2019) only exploited the symptom items. They first recognized symptom terms from dialogues and then classified their corresponding statuses; Zhang et al. (Zhang et al., 2020) proposed a deep neural matching model which is actually a multi-label text classification (MLTC) model. Each combination of item and status refers to an independent class label. Recently, Li et al. (Li et al., 2021b) modeled the extraction as a generation process, and developed a multi-granularity transformer which can effectively capture the interaction between role-enhanced crossturns and integrate representations of mixed granularity. Xia et al. (Xia et al., 2022) proposed a speaker-aware co-attention framework to address the intricate interactions among different utterances and the correlations between utterances and candidate items. All aforementioned methods disregard the relations among items as well as the relations between items and statuses.

In this paper, we propose a novel approach to model 1) the relations among items and 2) the relations between items and statuses in the DMIE task. Firstly, we adopt a pre-trained language model to learn representations for dialogues, items and statuses respectively; Secondly, we introduce

Table 1: Details of the labels in a DMIE dataset: MIEACL.

| Category | Item | Status |
|---|---|---|
| Symptom | Backache, Perspiration, Hiccups, Nausea, Cyanosis, Fever, Fatigue, Abdominal discomfort, · · · | Patient-Positive (appear/done/ normal) |
| Surgery | Interventional treatment, Radiofrequency ablation, Heart bypass surgery, Stent implantation, · · · | Patient-Negative (absent/not done/ abnormal) |
| Test | B-mode ultrasonography, CT examination, CT angiography, CDFI, Ultrasonography, MRI, Thyroid function test, Treadmill test, · · · | Doctor-Positive (diagnosed/suggest) Doctor-Negative (exclude/deprecated) |
| Other info | Sleep, Diet, Defecation, Smoking, Drinking, Mental condition, · · · | Unknown |

a heterogeneous graph to enrich the representations of items. In particular, we use the prior co-occurrences statistics and relationship information between items to guide the information transmission in the heterogeneous graph; Thirdly, we enrich the item representations with dialogue context and semantic information of status by introducing two attention based components, an *Item Updating with Dialogue-Aware Attention Module* and an *Item Refining with Status-Aware Module*. Finally, we use the joint representation which combines the item, status and dialogue representations as well as their mutual relations to predict the set of proper labels for this dialogue. We evaluate our approach on the MIEACL benchmark, and the results show that the proposed model outperforms all baselines. Further ablation study shows the effectiveness of our framework to model the diverse relationship among items, dialogues and statuses.

Overall, the main contributions are as follows:

- We propose a novel model for the DMIE, which introduces the interactions among medical items, incorporates the doctor-patient dialogue information to item representations and explores the item-status relationships.

- Rather than directly formulating DMIE as a multi-label text classification task, we design a status-aware item refining module for the DMIE task, which effectively captures the relationship between items and statuses.

- The experimental results on the MIEACL benchmark dataset show that our model significantly outperforms the state-of-the-art models, achieving a new SOTA performance.[2]

---

[2] https://github.com/DMIE-EMNLP2023/

## 2 Related Work

### 2.1 Dialogue Medical Information Extraction

Extracting medical information from doctor-patient dialogues has been receiving growing research interests. Shi et al. (2022) formulated dialogue medical information extraction as a slot filling task and proposed to extract symptoms from dialogues. Lin et al. (2019) detected symptom items as well their status in a pipeline fashion: first recognize symptom words from the dialogue and then classify them to predefined items and statuses. However, Lin et al. (2019) can only detect the items that are explicitly mentioned in the dialogue and ignore the implicitly mentioned ones. To identify implicitly mentioned symptoms, Du et al. (2019) proposed a Seq2Seq model to generate a sequence of symptoms and their statuses in an end-to-end manner. Recently, Zhang et al. (2020) utilized an attention based matching module to obtain category-specific representation and status-specific representation from the doctor-patient dialogues. The two acquired representations are further used to extract the targeted medical information. Li et al. (2021b) incorporated word-level information by using a Lattice-based encoder and a proposed relative position encoding method. Additionally, they proposed a role access controlled attention mechanism to incorporate utterance-level interaction information. Xia et al. (2022) proposed a speaker-aware dialogue encoder with multi-task learning, which incorporates the speaker's identity into the model. They also presented a co-attention fusion network to combine the information from different utterances and handle the intricate interactions among them, as well as the correlations between the utterances and candidate items.

Existing DMIE approaches treat each medical item independently. As a result, the correlations among medical items are ignored in modeling. Different from the above works, we employ a heterogeneous graph to directly model the correlations between medical items. Furthermore, we introduce an item updating with dialogue-aware attention module to infer implicit mention of medical items from dialogues.

### 2.2 Multi-Label Text Classification

The research of Multi-Label Text Classification (MLTC) focuses on two topics: document representation learning and label correlation learning. For a better understanding of relations between text and different labels, researchers began to exploit the knowledge of label correlations. Seq2emo (Huang et al., 2021) used a bi-directional LSTM (Hochreiter and Schmidhuber, 1997) decoder to implicitly model the correlations among different emotions; CorNet (Xun et al., 2020) designed a computational unit which was actually an improved version of a fully connected layer to learn label correlations, enhancing raw label predictions with correlation knowledge and output augmented label predictions; LDGN (Ma et al., 2021) employed dual Graph Convolution Network (GCN) (Kipf and Welling, 2017) to model the complete and adaptive interactions among labels based on the statistics of label co-occurrence, and dynamically reconstruct the graph in a joint way; (Zhang et al., 2021) designed a multi-task model, in which two additional label co-occurrence prediction tasks were proposed to enhance label relevance feedback.

In addition, there are also some other methods to improve model performance. For example, LightXML (Jiang et al., 2021) introduced a probabilistic label tree based clustering layer and a fully connected dynamic negative sampling layer on text representations to improve accuracy while controlling computational complexity and model sizes. For other NLP tasks like slot filling, slot correlations is learned with a self-attention mechanism (Ye et al., 2021). However, slot filling aims to extract the slots to fill in parameters of the user's query which is different from the targeted MIE task in this paper (classify dialog into predefined item status combinations).

Although DMIE could be treated as a special multi-label text classification task, the aforementioned methods are not able to process status information, which is not included in the ordinary MLTC datasets. Thus, we design an item refining with status-aware attention module to learn the cross attention between statuses and items to generate the status enhanced item representation.

## 3 Approach

### 3.1 Problem Definition

Given a dialogue with multiple consecutive turns of conversations, the objective of the DMIE task is to detect all predefined items, with their corresponding categories and mentioning statuses. Following (Zhang et al., 2020), we formulate the DMIE task as follows.

Given a dialogue $D$, we need to predict the labels

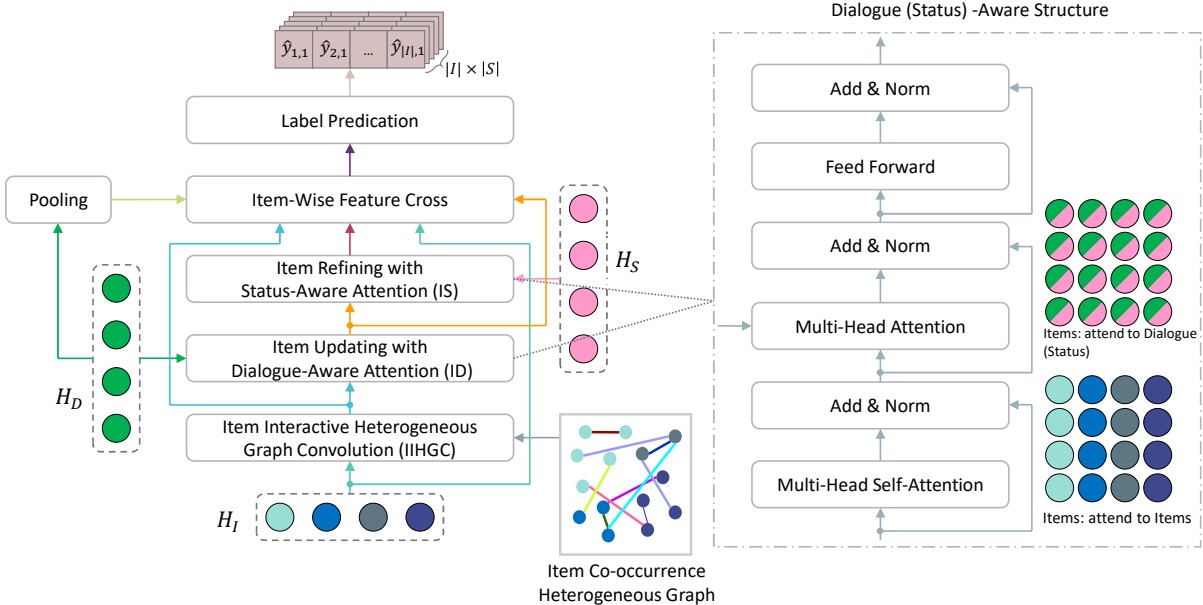

Figure 2: The workflow of the proposed model, which consists of three main components: Item Interactive Heterogeneous Graph Convolution (IIHGC) module, Item Updating with Dialogue-Aware Attention (ID) module and Item Refining with Status-Aware (IS) module.

$\{y_{i,j}\} \in \{0,1\}^{|I| \times |S|}$, where $I$ and $S$ represent the item set and status set, respectively, and $y_{i,j}$ denotes whether the item $i \in I$ with status $j \in S$ is mentioned in this dialogue. Here $|I|$ is the total number of predefined items, and $|S| = 5$ is the total number of statuses, i.e., *Patient-Positive*, *Patient-Negative*, *Doctor-Positive*, *Doctor-Negative* and *Unknown*. These items can be grouped with a set of $C$ categories, with a mapping $\tau(\cdot)$ where $\tau(i) \in C$ assigns the corresponding category, i.e., Symptom, Surgery, Test or Other info to the item $i$.

## 3.2 Dialogue, Item and Status Encoder

We denote a dialogue $D = \{u_1, u_2, \ldots, u_n\}$ with $n$ utterances, where $u_i = \{w_1^i, w_2^i, \ldots, w_{m^i}^i\}$, and $w_j^i$ is the $j$-th token in $i$-th utterance in the dialogue. We treat $D$ as a long sequence ($|D|$ tokens, $|D| = \sum_i^n m^i$ ) and feed it to the pre-trained language model to generate the contextualized representations $H_D = [h_1, \ldots, h_{|D|}], H_D \in \mathbb{R}^{|D| \times d}$, where $h_i \in \mathbb{R}^d$ is the contextualized representation of the $i$-th token with a dimension of $d$.

For pre-defined items and statuses, we feed their text descriptions into a pre-trained language model then use the mean-pooling of the last layer outputs to initialize their latent features $H_I = [h_1, \ldots, h_{|I|}], H_I \in \mathbb{R}^{|I| \times d}$ and $H_S = [h_1, \ldots, h_{|S|}], H_S \in \mathbb{R}^{|S| \times d}$.

## 3.3 Item Interactive Heterogeneous Graph Convolution

Items in DMIE tasks belong to four categories: *Symptom*, *Surgery*, *Test* and *Other info*, and the prior information between item pairs can be defined by the relationships between their categories. In Figure 1, when the patient said that he/she had the symptom of *frequent ventricular premature beats*, the doctor advised him to perform *electrocardiograms* testing and *radiofrequency ablation* surgery. As a result, we could observe a *"Symptom-Test"* relationship between *frequent ventricular premature beats* and *electrocardiograms*, and a *"Symptom-Surgery"* relationship between *frequent ventricular premature beats* and *radiofrequency ablation*. Such co-occurrence relationship between items is one prior knowledge that can be used in the DMIE task. As shown in Figure 2, we designed an Item Interactive Heterogeneous Graph Convolution (IIHGC) module to explicitly model the correlation among items.

### 3.3.1 Graph Construction

We use an undirected heterogeneous graph to model item correlations with different types of nodes and edges. We establish the edges between items based on their categories and the frequency of their co-occurrence in the corpus. Formally, our graph is denoted as $\mathcal{G} = (\mathcal{V}, \mathcal{E})$. Each node $v \in \mathcal{V}$ represents an item with a corresponding

category as $\tau(v) \in C$. Each edge $e = (v_i, v_j) \in \mathcal{E}$ represents a co-occurrence between items $v_i$ and $v_j$ from *different* categories. For example, the co-occurrence of *frequent ventricular premature beats* and *electrocardiogram* in Figure 1 gives an edge $\langle$"*Frequent Ventricular Premature Beats*", "*Symptom-Test*", "*Electrocardiogram*"$\rangle$ in our heterogeneous graph. In addition, the co-occurrence patterns between item pairs obtained from training data could contain noisy information. To reduce noise, we remove all edges with the number of co-occurrences less than four.

### 3.3.2 Heterogeneous Graph Convolution

After graph construction, we utilize a heterogeneous graph convolution module to model the item interactions, which is the IIHGC module in Figure 2. The IIHGC module takes the item representation $H_I = [h_1, \ldots, h_{|I|}]$ and the constructed graph as inputs and outputs high-level features $H_{HGC}$.

We first decompose the entire heterogeneous graph into several homogeneous subgraphs according to the types of edge relationships, and then apply independent Graph Attention Network (GAT) (Velickovic et al., 2018) on each subgraph to extract higher-level aggregated features. The graph attention network of each subgraph refines the representation of nodes by aggregating and updating information from its neighbors via multi-head attention mechanism. For a given edge from node $i$ to node $j$, the attention coefficient of the $k$-th head $\alpha_{ij}^k$ is calculated as follows:

$$\alpha_{ij}^k = \frac{\exp\left(\text{LeakyReLU}\left(a^\top[\mathbf{W}_1^k h_i \| \mathbf{W}_2^k h_j]\right)\right)}{\sum_{k \in \mathcal{N}_i} \exp\left(\text{LeakyReLU}\left(a^\top[\mathbf{W}_1^k h_i \| \mathbf{W}_2^k h_k]\right)\right)}$$

where $d'$ is the dimension of the head attention, $\mathbf{W}_1^k \in \mathbb{R}^{d' \times d}$, $\mathbf{W}_2^k \in \mathbb{R}^{d' \times d}$, $a \in \mathbb{R}^{2d'}$ are model parameters, $\mathcal{N}_i$ is the neighbors of node $i$ and $\|$ denotes the concatenation operation.

Next, the representation of the $k$-th head for node $i$ is obtained by linear combination of attention coefficients and corresponding features:

$$h_i^k = \sigma\left(\sum_{j \in \mathcal{N}_i} \alpha_{ij}^k \mathbf{W}^k h_j\right)$$

where $\sigma$ is the activation function.

After that, we get the final information-aggregated output of node $i$ in this subgraph by concatenating the generated $K$ independent attention heads outputs:

$$H_i = \|_{k=1}^K h_i^k$$

where $\|$ represents concatenation and $H_i$ has a dimension of $K \times d' = d$ features.

Finally, we aggregate messages from all subgraph with different relationships for each node:

$$H_{HGC_i} = \sum_{r=1}^{|R|} H_{r_i}$$

where $H_{r_i}$ is the representation of item $i$ calculated from the $r$-th subgraph and $|R|$ is the total number of relationship categories included in the heterogeneous graph.

In our setting, the final output of node $i$, $H_{HGC_i}$, has a dimension of $d$ features.

## 3.4 Item Updating with Dialogue-Aware Attention

In the DMIE task, each item corresponds to a normalized name. The content of the doctor-patient dialogue can help us to determine which item is mentioned, and its corresponding status. In this paper, we utilize item-dialogue cross attention mechanism to enrich the information of item representation for better classification, which is denoted as the Item Updating with Dialogue-Aware Attention (ID) module in Figure 2.

We obtain the dialogue representation $H_D$, item representation $H_I$ from the dialogue encoder and the item encoder, respectively. Then we update the item representation through the aforementioned Item Interactive Heterogeneous Graph module and obtain the representation $H_{HGC}$. Next, in the ID module, we use the attention mechanism similar to the Transformer block (Vaswani et al., 2017) to model the relation between the dialogue and items:

$$H'_{HGC} = \text{Add\&Norm}(H_{HGC}, \text{MultiHead}(H_{HGC}, H_{HGC}, H_{HGC})),$$
$$H_{ID}^1 = \text{Add\&Norm}(H'_{HGC}, \text{MultiHead}(H'_{HGC}, H_D, H_D)),$$
$$H_{ID} = \text{Add\&Norm}(H_{ID}^1, \text{FFN}(H_{ID}^1))$$

where Add&Norm, MultiHead and FFN are the standard Transformer operations (Vaswani et al., 2017). Since there is no autoregressive or causal relationship between items, we do not use attention masks in the self attention layer.

## 3.5 Item Refining with Status-Aware Attention

One major difference between the DMIE task and the conventional multi-label text classification task is that it needs to assign a status to each item. Therefore, in this paper, we design an Item Refining with Status-Aware Attention (IS) module in Figure 2,

to enrich the representation of items. The IS module uses a mechanism similar to the ID module to refine the representation of items:

$$H'_{ID} = \text{Add\&Norm}(H_{ID}, \text{MultiHead}(H_{ID}, H_{ID}, H_{ID})),$$
$$H^1_{IS} = \text{Add\&Norm}(H'_{ID}, \text{MultiHead}(H'_{ID}, H_S, H_S)),$$
$$H_{IS} = \text{Add\&Norm}(H^1_{IS}, \text{FFN}(H^1_{IS}))$$

where $H_{IS}$ is the refined status-aware item representation. Following the ID module, we also discard the attention mask in the self attention layer in the IS module.

### 3.6 Label Prediction and Optimization Objectives

After the above procedures, we obtain the pooled dialogue representation $h_d = \text{MeanPooling}(H_D)$, item embedding $H_I$, item-graph interacted representation $H_{HGC}$, dialogue-aware item representation $H_{ID}$ and status-aware item representation $H_{IS}$. We predict the $|S|$-dimension status of item $i$ by:

$$\hat{y}_i = \sigma\Big(\mathbf{W}_c\big[h_d \| f(h_d, H_{I_i}) \| f(h_d, H_{HGC_i})$$
$$\| f(h_d, H_{ID_i}) \| f(h_d, H_{IS_i})\big]\Big),$$

$$f(h_1, h_2) = [h_2 \| \text{abs}(h_1 - h_2)]$$

where $\sigma$ is the sigmoid activation function and $\mathbf{W}_c \in \mathbb{R}^{|S| \times 9d}$ are the model parameters used for classification.

The proposed model is trained with the binary cross entropy loss:

$$\mathcal{L} = \sum_{i \in |I|} \sum_{j \in |S|} y_{i,j} \log(\hat{y}_{i,j}) + (1 - y_{i,j}) \log(1 - \hat{y}_{i,j})$$

## 4 Experiments

### 4.1 Dataset and Preprocess

We validate our model on the MIEACL benchmark from an online medical consultation website and proposed by (Zhang et al., 2020). (Zhang et al., 2020) divided each dialogue into multiple parts using a sliding window, the sliding step is 1 and the window size is 5. In this manner, a dialogue with $k$ conversation turns will be split into $k$ windows of size 5. In total, the 1,120 dialogues in MIEACL are divided into 18,212 windows.

### 4.2 Evaluation Metrics and Settings

Following the settings of (Zhang et al., 2020), we use the MIE-Macro Precision, MIE-Macro Recall and MIE-Macro F1 score as our main evaluation metrics, which produce the metrics over the average performance of each sliding window. It is worth mentioning that, different from ordinary Macro-Average metrics, the MIE-Macro Precision and MIE-Macro Recall is set to 1 by (Zhang et al., 2020) if the prediction is empty on a window with no labels. To reduce the impact of samples with no labels, we additionally use Micro-Average metrics for further comparison. For a fair comparison, we apply the same train/dev/test dataset split as (Zhang et al., 2020), and select the best performing model in the validation set for testing.

We utilize the Chinese Roberta (Cui et al., 2019) as the encoder in our model, which is a base version of the pre-trained Chinese RoBERTa-wwm-ext. The ID, IIHGC, and IS have the same dropout rate, attention heads, and hidden layer dimensions as the original RoBERTa model, which are 0.1, 12, and 768 respectively. We employ the AdamW optimizer for training our model, with a learning rate of 2e-5 for the parameters in the RoBERTa encoder and 2e-4 for the rest of the model. We incorporate an early stop mechanism to avoid overfitting, whereby the training is stopped if there is no improvement in the MIE-Macro F1 or MIE-Micro F1 on the validation set over 10 epochs. Running an experiment with these settings will roughly take 9 hours on a single NVidia V100 GPU.

### 4.3 Baseline Models

To demonstrate the effectiveness of our proposed model, we compare it with seven strong baselines:

- **RoBERTa$_{\text{CLS}}$**: RoBERTa$_{\text{CLS}}$ is a basic classification model which employs the top-level representation $h_{\text{CLS}}$ to perform medical information extraction.

- **LightXML** (Jiang et al., 2021): LightXML adopts end-to-end training, label clustering and dynamic negative label sampling to improve model performance.

- **CorNet** (Xun et al., 2020): CorNet adds an extra module to learn label correlations to enhance raw label predictions and output augmented label predictions.

Table 2: Results(%) on the MIEACL dataset, where P stands for Precision, R stands for Recall, and F1 is the standard F1 metric. IIHGC+ID+IS is our proposed model, which can be observed to outperform over all baselines. [†] indicates the results from original paper.

| Methods | MIE-Macro Metric | | | Micro Metric | | |
| --- | --- | --- | --- | --- | --- | --- |
| | P | R | F1 | P | R | F1 |
| RoBERTa$_{CLS}$ | 77.80 | 75.40 | 75.35 | 75.77 | 69.90 | 72.72 |
| LightXML | 74.72 | 70.81 | 71.38 | 72.45 | 62.17 | 66.92 |
| CorNet | 73.07 | 71.43 | 71.00 | 67.72 | 63.72 | 65.66 |
| MIE | 69.59 | 65.89 | 66.05 | 69.43 | 61.77 | 65.38 |
| SAFE[†] | 72.59 | 73.86 | 73.22 | - | - | - |
| MGT[†] | 75.30 | 71.70 | 72.70 | - | - | - |
| IIHGC+ID+IS | **82.77** | **83.60** | **82.54** | **81.35** | **81.89** | **81.62** |

- **MIE** (Zhang et al., 2020): MIE focuses on learning an improving representation using a deep matching architecture that takes into account dialogue-turn interactions to capture the category-item pair information and the status information, and feed them both into a classifier for text classification.

- **SAFE** (Xia et al., 2022): SAFE is a speaker-aware model that employs a co-attention fusion technique with multitask learning and graph networks.

- **MGT** (Li et al., 2021b): The MGT model proposes a Multi-Granularity Transformer to fully capture the interaction between role-enhanced cross-turns and integrate mixed granularity representations.

## 4.4 Results and Discussion

We report the experimental results of all comparing algorithms on the MIEACL dataset in Table 2.

Our approach outperforms previous work (Zhang et al., 2020) on the DMIE task. The combination of IIHGC, ID, and IS achieves 82.54% MIE-Macro F1 and 81.62% Micro F1 scores, outperforming the previous work by 16.49% and 16.24% respectively. The performance gain is mainly due to the introduction of internal correlations between items and statuses. Our model not only introduces prior information between items, but also uses the self-attention mechanism to effectively update and refine item representations. Please note that since we need to calculate the Micro metrics in addition to the Marco metrics, for consistency consideration, we re-run the code of (Zhang et al., 2020) on the benchmark data set instead of directly citing the numbers reported (Zhang et al., 2020).

Table 3: Quantitative analysis(%) of our model with different variants.

| Variants | IIHGC | ID | IS | MIE-Macro Metric | | | Micro Metric | | |
| --- | --- | --- | --- | --- | --- | --- | --- | --- | --- |
| | | | | P | R | F1 | P | R | F1 |
| 1 | | | | 79.30 | 78.20 | 77.57 | 74.52 | 74.38 | 74.45 |
| 2 | √ | | | 79.84 | 79.05 | 78.31 | 76.25 | 73.86 | 75.04 |
| 3 | | √ | | 82.40 | 82.73 | 81.82 | 81.08 | 81.08 | 81.08 |
| 4 | | | √ | 78.97 | 79.36 | 78.01 | 75.12 | 74.86 | 74.99 |
| 5 | √ | √ | | 82.19 | 83.19 | 81.94 | 80.43 | 81.46 | 80.94 |
| Full | √ | √ | √ | **82.77** | **83.60** | **82.54** | **81.35** | **81.89** | **81.62** |

We can also see that our proposed method outperforms the conventional Multi-Label Text Classification (MLTC) approaches (Jiang et al., 2021; Xun et al., 2020). This is mainly due to that the MLTC models do not consider the complex relationships between the conversation contents and items, as well as the corresponding relationships between items and status. Our model outperforms these models by more than 11% and 14% on MIE-Macro F1 score and Micro F1 score, respectively.

To confirm that the improvement does not merely come from the adoption of pre-trained language model, we compare our model with RoBERTa$_{CLS}$. The performance of RoBERTa$_{CLS}$ shows that the adoption of a high-quality pre-trained model is helpful for the DMIE task. Nevertheless, our method surpasses this baseline by additional 7.19% and 8.9% on MIE-Macro F1 score and Micro F1 score, respectively, showing the effectiveness of the introduction of our IIHGC, ID and IS modules.

## 4.5 Ablation Study

### 4.5.1 Results of Different Model Variants

We conduct an ablation study by gradually stripping components to examine the effectiveness of each component in our full model (IIHGC+ID+IS). The experiment results are illustrated in Table 3. Variant$_1$ is the most basic variant which only uses the dialogue representation and item embeddings for label prediction. Benefiting from the introduction of item information, the model Variant$_1$ has effectively improved the text classification based approach (RoBERTa$_{CLS}$). However, due to the lack of medical item correlation information, the reference relationship between dialogue and item, and the corresponding relationship between item and status, Variant$_1$ gets the worst performance among all variants. The performances of Variant$_{[2-4]}$ are better than Variant$_1$, which proves the effectiveness of the introduced IIHGC, ID and IS components. A single ID module brings the largest improvement

Table 4: Results(%) on the MIEACL Dataset with Different GNN models

| Methods | MIE-Macro Metric | | | Micro Metric | | |
|---|---|---|---|---|---|---|
| | P | R | F1 | P | R | F1 |
| MIE | 69.59 | 65.89 | 66.05 | 69.43 | 61.77 | 65.38 |
| IIHGC with (Velickovic et al., 2018) | 82.77 | 83.60 | 82.54 | 81.35 | 81.89 | 81.62 |
| IIHGC with (Corso et al., 2020) | 82.56 | 83.54 | 82.35 | 81.43 | 81.40 | 81.41 |
| IIHGC with (Chen et al., 2020) | 82.42 | 82.65 | 81.74 | 81.72 | 81.24 | 81.48 |
| IIHGC with (Li et al., 2021a) | 81.15 | 83.76 | 81.67 | 81.15 | 81.69 | 81.41 |

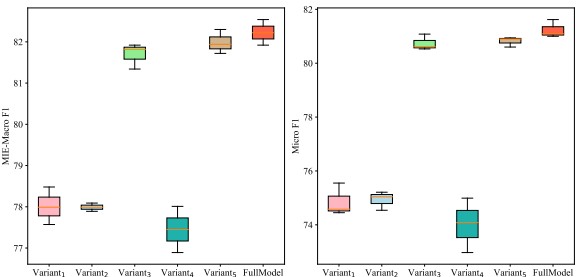

Figure 3: Model performance distribution under different random seeds.

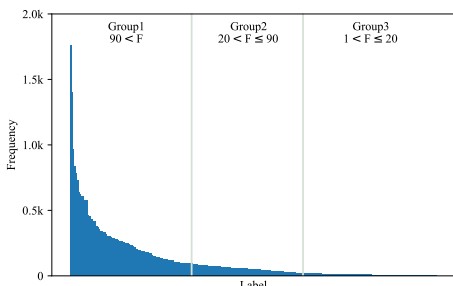

Figure 4: The distribution of medical item+status label frequency on the MIEACL benchmark dataset.

compared to the baseline variant. With the combination of IIHGC, ID and IS modules, our full model is able to obtain improvements of 4.97% on the MIE-Macro F1 score and 7.17% on the Micro F1 score over Variant$_1$. Overall, removal of any introduced components will cause the classification performance to drop, which confirms the contribution of each component in the final model and proves their effectiveness.

In Section 3.3, we use Graph Attention Network (GAT) introduced in (Velickovic et al., 2018) to model the relations among items. There are also other graph neural network approaches like (Corso et al., 2020; Chen et al., 2020; Li et al., 2021a). Here we compare the performance of the IIHGC equipping with different GNN models. As shown in Table 4, we can find that equipping with any of these four graph models, the proposed method consistently outperforms SOTA works.

We further explore the sensitivity of the model to random seed setting, and the results are presented in Figure 3. Here five different random seeds were used. We find that the full model performance remains stable under different random seeds and is consistently better than other model variants. To sum up, our proposed method can effectively improve the performance and is robust in training.

### 4.5.2 Results of Different Label Frequencies

Now we study the performance on labels with varied frequencies. Here a label means the combina-

tion of an item and its specified status. As shown in Figure 4, the label distribution in the actual medical environment is naturally long tailed. Therefore, such long-tail bias may have impact on the performance of the model and its practicability. Here we divide all the labels into three groups according to their frequencies in the training set to explore the performance of the model: the major group (Group 1, $F > 90$), the mid-frequency group (Group 2, $20 < F \leq 90$), and the long-tail group (Group 3, $1 \leq F \leq 20$). We calculate the accuracy of each label and add it into its corresponding group average accuracy. As shown in Table 5, we find that the performance of all methods gradually declines when the the label frequency becomes smaller, but the performance of the full model is constantly higher than all variants and decreases in a slower rate. In Group 3, we can see that the full model achieves 28.65% improvement compared with the baseline variant model. We speculate the reason is due to the incapability of the baseline variants to understand the semantics of labels, which restricts their performance. The full model not only generates fine-grained item representations for different dialogues, but also discovers the implicit semantics of long-tail labels.

### 4.5.3 Results of Different Dialogue Lengths

Medical conversations varied in length which is usually determined by the complexity of patients' diseases. The MIEACL dataset adopts a conven-

Table 5: Labels with different frequencies. Groups 1, 2, and 3 correspond to the divisions in Figure 4.

| Variants | Component | | | Group | | |
| --- | --- | --- | --- | --- | --- | --- |
| | IIHGC | ID | IS | 1 | 2 | 3 |
| 1 | | | | 80.20 | 43.71 | 27.03 |
| 2 | ✓ | | | 80.87 | 44.70 | 27.57 |
| 3 | | ✓ | | 85.70 | 61.81 | 53.51 |
| 4 | | | ✓ | 82.87 | 48.79 | 31.35 |
| 5 | ✓ | ✓ | | 85.41 | **66.00** | 52.43 |
| Full Model | ✓ | ✓ | ✓ | **85.91** | 65.12 | **55.68** |

Table 6: F1 (%) of dialogues with different lengths.

| Variants | Component | | | Length Group | |
| --- | --- | --- | --- | --- | --- |
| | IIHGC | ID | IS | L ≤ 300 | L > 300 |
| 1 | | | | 77.54 | 69.14 |
| 2 | ✓ | | | 77.84 | 78.49 |
| 3 | | ✓ | | 81.62 | 87.63 |
| 4 | | | ✓ | 76.79 | 77.17 |
| 5 | ✓ | ✓ | | 81.76 | 85.83 |
| Full Model | ✓ | ✓ | ✓ | **82.34** | **87.69** |

tional data annotation scheme by dividing the dialogue into multiple windows to predict the item mentions and their statuses at the window level. As a result, the longer conversations will be divided into more instances which could bring in data bias in training. In this part, we divide the test set into two groups according to the length of the dialog to compare the model variants: length less than or equal to 300 (Group 1) and length greater than 300 (Group 2). The experimental results are shown in the Table 6. Our full model still significantly outperforms other model variants in both groups. It is worth mentioning that only the performance of the Variant$_1$ decreases as the input length increases. The performance of the remaining variants on Group 2 surpassed their performance on Group 1. This is mainly because longer dialogues contain more information. Though the baseline model Variant$_1$ cannot capture this information well, the other variants with one or more of our proposed components effectively extract these key information from different perspectives. The empirical results clearly demonstrate the superiority of our model on longer dialogues, which contain richer information.

### 4.5.4 Comparison with LLM

We also evaluate the performance of GPT-3.5-turbo and GPT-4 on the DMIE task. Here we design a specific prompt and adopt one-shot learning setting. The prompt template can be found in Figure 5. The experimental results are shown in Table 7. We can find that GPT-4 outperformed GPT-3.5-turbo, but

Table 7: Results(%) on the MIEACL dataset, Comparison with LLM approaches

| Methods | MIE-Macro Metric | | | Micro Metric | | |
| --- | --- | --- | --- | --- | --- | --- |
| | P | R | F1 | P | R | F1 |
| GPT-3.5-Turbo | 46.14 | 40.46 | 43.11 | 39.37 | 29.96 | 34.03 |
| GPT-4 | 54.92 | 62.18 | 58.33 | 44.36 | 56.72 | 49.78 |
| IIHGC+ID+IS | **82.77** | **83.60** | **82.54** | **81.35** | **81.89** | **81.62** |

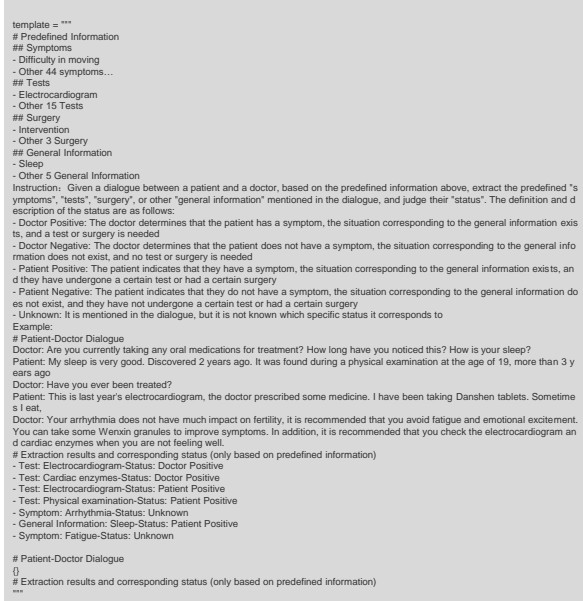

Figure 5: The One-Shot Prompt Template of the DMIE Task for GPT-3.5 and GPT-4

both underperformed SOTA MIE works and the proposed IIHGC by a large margin. This is may be due to, for this particular MIE task, the supervised models are more suitable than the general LLM approaches.

## 5 Conclusions

We introduce a heterogeneous item graph to model item correlations, as well as two attention based modules to learn a dialogue-status enriched joint representation for Dialogue Medical Information Extraction (DMIE). Instead of formulating DMIE as an ordinary multi-label text classification problem, we consider the item-status relationship and model this relationship explicitly. Extensive experiments demonstrated that each component of our proposed model could bring performance gains to the baseline model, and that their combination further improved the result and achieved the state-of-the-art performance on the MIEACL benchmark. We also evaluate the performance of GPT-3.5-turbo and GPT-4. In future, we will attempt to include textual names of items and status in modeling.

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

# A  Limitations

This paper aims to extract predefined medical items and status from doctor-patient dialogues. One major limitation is that our approach needs pre-defined medical items and status, this is also the common limitation for all DMIE approaches. Currently the pre-defined items and status cover common diseases and symptoms, when applying to relatively rare diseases, new medical items and status labels need to be introduced, and new training data needs to be labeled.

## B Ethics Statement

In this paper, we target the Dialogue Medical Information Extraction task. Although the proposed model achieves state-of-the-art results, it aims to assist doctors in online diagnosis and the construction of Electronic Medical Records (EMRs) rather replace the doctors. Our model can relieve the doctors from the tedious, time-consuming and error prone tasks of labelling relevant medical items and status, but inexperienced doctor may be over-reliant on it. Therefore, it is necessary to introduce additional quality controls to avoid the abuse of our model.