# OpenReview forum: "Dialogue Medical Information Extraction with Medical-Item Graph and Dialogue-Status Enriched Representation"
_EMNLP/2023/Conference — EMNLP 2023 Findings_

### Official Review · Reviewer_hgH8 · 2023-07-26

**Soundness:** 3

**Excitement:**

3: Ambivalent: It has merits (e.g., it reports state-of-the-art results, the idea is nice), but there are key weaknesses (e.g., it describes incremental work), and it can significantly benefit from another round of revision. However, I won't object to accepting it if my co-reviewers champion it.

**Missing References:**

- I did not find.

**Paper Topic And Main Contributions:**

This work reports some experiments conducted on the DMIE dataset (Zhang et al. 2020) for a medical information extraction task using Chinese doctor-patient dialogues. The authors present a neural-network-based model using the correlation between each medical entity pair (modeled as a relationship between nodes in a graph, IIHGC module); the relation between each dialogue representation and each item representation; and the relation between each item and its status. A representation of each dialogue is obtained by applying a mean pooling technique on the aggregation of these information types. Then a sigmoid function is applied to train and optimize the model parameters. The authors compare their architecture with previous models: MIE (Zhang et al. 2020), LightXML (Jiang et al 2021), CorNet (Xun et al. 2020), Roberta CLS, MGT (Li et al., 2021) and SAFE (Xia et al., 2022). The evaluation metrics were obtained by averaging results in each sliding window of 5 turns. The authors's results outperform those obtained with the other models, and include an ablation analysis. Experiments are also repeated using different frequencies (f) of labels in the dataset (f>90, 20<f<90, 1<f<20) and with longer (>300 turns) or shorter dialogues (<300 turns).

**Questions For The Authors:**

- Table 1: How are disorders or pharmacological substances modeled? Are they included in the "Other info" category?
- L. 530ff: How many random seeds were used to explore the variability of results?
- L. 437ff (sect. 4.3): more details about the hyperparameters are missing, and also the GPU type or number of training hours. Moreover, given that this is a sophisticated model that needs long pretraining, the authors could have brought out the discussion about its advantages compared to other frameworks for the dialogue task; especially, when large volumes of data are hard to obtain.


**Reasons To Accept:**

- A new architecture to extract medical information in dialogue data. Dialogue context and the correlation between medical entities is incorporated.
- An extensive evaluation including ablation analysis and experiments with variants of dialogue length and label frequencies. I recommend the authors to include conduct a qualitative analysis or an error analysis in future work; this could shed more light to understand the model's results.
- Overall, I think this contribution is well writen, reports extensive work and could be of interest of the conference audience.


**Reasons To Reject:**

- Some technical aspects about how the dialogue representation is obtained are missing. I could not find a explicit statement to explain whether the graph from the IIHGC model is obtained for each 5-turn window, and trained separately, or whether the graph is obtained once from the full DMIE dataset.
- Not much information is provided about how the term variation is processed. Medical term variation/normalization modules could increase the results. Since data from doctor-patient encounters are very scarce, these components could be explored as a complement to neural network frameworks.


**Reproducibility:**

4: Could mostly reproduce the results, but there may be some variation because of sample variance or minor variations in their interpretation of the protocol or method.

**Reviewer Confidence:**

3: Pretty sure, but there's a chance I missed something. Although I have a good feel for this area in general, I did not carefully check the paper's details, e.g., the math, experimental design, or novelty.

**Typos Grammar Style And Presentation Improvements:**

- L. 414: please, at the beginning of the sentence, change "(Zhang et al., 2020)" to "Zhang et al. (2020)"; use Latex "citet{}"

---

> ### Author Rebuttal · Authors · 2023-08-29
>
> We thank the reviewer hgH8 for the valuable comments. Please kindly find the point-to-point responses below.
>
> **Q1**: Table 1: How are disorders or pharmacological substances modeled? Are they included in the "Other info" category?
>
> **A1**: For the targeted MIE task, as shown in Table 1, four types of items and five kinds of status are introduced. The items as well as the status are predefined. Our task is to determine whether a combination of item and status can be infered from current input dialogue. "Other info" is a separate category, which includes sleep, diet, mental state, bowel movements, smoking, and drinking. The complete list of items can be found below.
>
> ```
> ## Symptoms
> - Difficulty in moving
> - Tremors
> - Palpitations
> - Back pain
> - Dizziness
> - Burping
> - Abdominal discomfort
> - High blood pressure
> - High blood sugar
> - Difficulty breathing
> - Chest tightness
> - High blood lipids
> - Nausea
> - Vomiting
> - Chest pain
> - Fatigue
> - Sweating
> - Fever
> - Shock
> - Fainting
> - Cold
> - Cough
> - Runny nose
> - Headache
> - Stomach discomfort
> - Stiffness
> - Cyanosis
> - Diabetes
> - Anemia
> - Edema
> - Angina
> - Hyperthyroidism
> - Premature beat
> - Irregular heartbeat
> - Atrial septal defect
> - Atrial fibrillation
> - Heart failure
> - Myocardial infarction
> - Congenital heart disease
> - Myocardial ischemia
> - Ventricular septal defect
> - Myocarditis
> - Coronary heart disease
> - Cardiomyopathy
> - Cardiac hypertrophy
> ## Tests
> - Electrocardiogram
> - Color ultrasound
> - Cardiac enzymes
> - Physical examination
> - Angiography
> - Ultrasound
> - CT
> - Complete blood count
> - Thyroid function
> - Chest X-ray
> - B-ultrasound
> - Kidney function
> - Treadmill
> - CTA
> - Blood pressure measurement
> - MRI
> ## Surgery
> - Intervention
> - Radiofrequency ablation
> - Bypass
> - Stent
> ## Others
> - Sleep
> - Diet
> - Mental state
> - Bowel movements
> - Smoking
> - Drinking
> ```
>
>
> **Q2**: L. 530ff: How many random seeds were used to explore the variability of results?
>
> **A2**: Five different random seeds were used.
>
>
>
> **Q3**: L. 437ff (sect. 4.3): more details about the hyperparameters are missing, and also the GPU type or number of training hours. Moreover, given that this is a sophisticated model that needs long pretraining, the authors could have brought out the discussion about its advantages compared to other frameworks for the dialogue task; especially, when large volumes of data are hard to obtain.
>
>
> **A3**: We utilize the hfl/chinese-roberta-wwm-ext as the encoder in our model, which is a base version of the pre-trained Chinese RoBERTa-wwm-ext. The ID, IIHGC, and IS have the same dropout rate, attention heads, and hidden layer dimensions as the original RoBERTa model, which are 0.1, 12, and 768 respectively. We employ the AdamW optimizer for training our model, with a learning rate of 2e-5 for the parameters in the RoBERTa encoder and 2e-4 for the rest of the model. We incorporate an early stop mechanism to avoid overfitting, whereby the training is stopped if there is no improvement in the MIE-Macro F1 or MIE-Micro F1 on the validation set over 10 epochs. Running an experiment with these settings will roughly take 9 hours on a single Nvidia V100 GPU.
>
>
> **Q4**: Some technical aspects about how the dialogue representation is obtained are missing. I could not find a explicit statement to explain whether the graph from the IIHGC model is obtained for each 5-turn window, and trained separately, or whether the graph is obtained once from the full DMIE dataset.?
>
> **A4**: The graph is obtained once from the training set of the DMIE dataset. Section 3.1.1 of the paper discusses the steps to construct the graph. Each node v represents an item with its corresponding category, and the graph's edges are inferred from the co-occurrence relationships among items. In detail, if the co-occurrence of two items (within the same dialogue) exceeds four times in the training set, they are considered to be connected by an edge. The type of the edge is determined by the types of two connected nodes.
>
>
> **Q5**: Not much information is provided about how the term variation is processed. Medical term variation/normalization modules could increase the results. Since data from doctor-patient encounters are very scarce, these components could be explored as a complement to neural network frameworks.?
>
> **A5**:  Thanks for your great advice. We totally agree that medical term variation/normalization modules could increase the performance of MIE task. In this paper, we use the item refining module to handle term variations between doctor-patient conversations. We first encode the representation of items using a language model, and introduce the relationship graph between items to further enrich the representation of items. Then, the representations of these items and conversations are fed into a cross attention network to perceive the mention of items. The acquired representations of items may help to provide term variation/normalization functions and allievate the data scarce problem.

---

### Official Review · Reviewer_DNoq · 2023-08-05

**Soundness:** 4

**Excitement:**

4: Strong: This paper deepens the understanding of some phenomenon or lowers the barriers to an existing research direction.

**Paper Topic And Main Contributions:**

In this paper, the authors describe their system for classifying dialogue medical information including one of several overall categories (symptom, surgery, test, other info), more detailed items (e.g. stent implantation), and status (e.g. patient-positive). Their system outperforms several baselines and contains performance analysis related to most frequent categories.

In general the paper is well written, including relevant descriptions of the problem, related works, baselines, and performance analysis.

The system description seems like it provides necessary information; this combined with code will allow users to understand the system.

Minor Comments
-------------------
4.1 line 413-414 has a repeat reference

Looking at the results of this works’ cited paper (https://aclanthology.org/2020.acl-main.576.pdf) which is also the dataset paper, there was breakdown by both window/dialogue level as well as by category/item (and full). Is the cited Zhang 2020 paper table 4 score of 66.40 F1 supposed to be correlate somewhere with this papers’ Table 3 MIE Micro F1 65.38? (if yes, perhaps put a small footnote on why it’s different would be good)


**Reasons To Accept:**

- This is an important problem
- Meaningful improvement over baselines

**Reasons To Reject:**

- Comparison with newer large pretrained language models would be nice

**Reproducibility:**

4: Could mostly reproduce the results, but there may be some variation because of sample variance or minor variations in their interpretation of the protocol or method.

**Reviewer Confidence:**

3: Pretty sure, but there's a chance I missed something. Although I have a good feel for this area in general, I did not carefully check the paper's details, e.g., the math, experimental design, or novelty.

---

> ### Author Rebuttal · Authors · 2023-08-29
>
> We thank the reviewer DNoq for the valuable comments. Please kindly find the point-to-point responses below.
>
>
> **Q1**: Comparison with newer large pretrained language models would be nice
>
> **A1**: Following your great advice, we conduct experiments on gpt3.5-turbo and gpt4. We design a prompt for the MIE task and use one-shot learning setting. The experimental results and the prompt can be found below.
>
>
> |     **Model**     | **Macro Metric**| | | **Micro Metric**| | |
> | :----------- | :-----------: | :-----------: | :-----------: | :-----------: | :-----------: | :-----------: |
> |        | Precision| Recall | F1 | Precision| Recall | F1 |
> |gpt3.5-turbo	|46.14|	40.46|	43.11|	39.37|	29.96	|34.03|
> |gpt4|	54.92|	62.18|	58.33|	44.36	|56.72|	49.78|
> | IIHGC    | 82.77 | 83.60 | 82.54 | 81.35 | 81.89 | 81.62 |
>
> We can find that gpt4 outperformed gpt3.5-turbo, but both underperform SOTA works and the proposed IIHGC by a large margin. This is may be due to for this particular MIE task, the supervised models are more suitable than the general LLM approaches.
>
> Below is the template of prompt (one-shot setting):
>
> ```
> template = """
> # Predefined Information
> ## Symptoms
> - Difficulty in moving
> - Tremors
> - Palpitations
> - Back pain
> - Dizziness
> - Burping
> - Abdominal discomfort
> - High blood pressure
> - High blood sugar
> - Difficulty breathing
> - Chest tightness
> - High blood lipids
> - Nausea
> - Vomiting
> - Chest pain
> - Fatigue
> - Sweating
> - Fever
> - Shock
> - Fainting
> - Cold
> - Cough
> - Runny nose
> - Headache
> - Stomach discomfort
> - Stiffness
> - Cyanosis
> - Diabetes
> - Anemia
> - Edema
> - Angina
> - Hyperthyroidism
> - Premature beat
> - Irregular heartbeat
> - Atrial septal defect
> - Atrial fibrillation
> - Heart failure
> - Myocardial infarction
> - Congenital heart disease
> - Myocardial ischemia
> - Ventricular septal defect
> - Myocarditis
> - Coronary heart disease
> - Cardiomyopathy
> - Cardiac hypertrophy
> ## Tests
> - Electrocardiogram
> - Color ultrasound
> - Cardiac enzymes
> - Physical examination
> - Angiography
> - Ultrasound
> - CT
> - Complete blood count
> - Thyroid function
> - Chest X-ray
> - B-ultrasound
> - Kidney function
> - Treadmill
> - CTA
> - Blood pressure measurement
> - MRI
> ## Surgery
> - Intervention
> - Radiofrequency ablation
> - Bypass
> - Stent
> ## General Information
> - Sleep
> - Diet
> - Mental state
> - Bowel movements
> - Smoking
> - Drinking
>
> Instruction：Given a dialogue between a patient and a doctor, based on the predefined information above, extract the predefined "symptoms", "tests", "surgery", or other "general information" mentioned in the dialogue, and judge their "status". The definition and description of the status are as follows:
> - Doctor Positive: The doctor determines that the patient has a symptom, the situation corresponding to the general information exists, and a test or surgery is needed
> - Doctor Negative: The doctor determines that the patient does not have a symptom, the situation corresponding to the general information does not exist, and no test or surgery is needed
> - Patient Positive: The patient indicates that they have a symptom, the situation corresponding to the general information exists, and they have undergone a certain test or had a certain surgery
> - Patient Negative: The patient indicates that they do not have a symptom, the situation corresponding to the general information does not exist, and they have not undergone a certain test or had a certain surgery
> - Unknown: It is mentioned in the dialogue, but it is not known which specific status it corresponds to
>
> Example:
> # Patient-Doctor Dialogue
> Doctor: Are you currently taking any oral medications for treatment? How long have you noticed this? How is your sleep?
> Patient: My sleep is very good. Discovered 2 years ago. It was found during a physical examination at the age of 19, more than 3 years ago
> Doctor: Have you ever been treated?
> Patient: This is last year's electrocardiogram, the doctor prescribed some medicine. I have been taking Danshen tablets. Sometimes I eat,
> Doctor: Your arrhythmia does not have much impact on fertility, it is recommended that you avoid fatigue and emotional excitement. You can take some Wenxin granules to improve symptoms. In addition, it is recommended that you check the electrocardiogram and cardiac enzymes when you are not feeling well.
> # Extraction results and corresponding status (only based on predefined information)
> - Test: Electrocardiogram-Status: Doctor Positive
> - Test: Cardiac enzymes-Status: Doctor Positive
> - Test: Electrocardiogram-Status: Patient Positive
> - Test: Physical examination-Status: Patient Positive
> - Symptom: Arrhythmia-Status: Unknown
> - General Information: Sleep-Status: Patient Positive
> - Symptom: Fatigue-Status: Unknown
>
> # Patient-Doctor Dialogue
> {}
> # Extraction results and corresponding status (only based on predefined information)
> """
> ```
>
>
> **Q2**:  Is the cited Zhang 2020 paper table 4 score of 66.40 F1 supposed to be correlate somewhere with this papers’ Table 3 MIE Micro F1 65.38?
>
> **A2**:  In Zhang 2020 paper table 4, the score of 66.40 F1 is the MIE-Marco metric which corresponds to 66.05 in Table 2 of this paper. Since we need to calculate the Micro metrics in addition to the Marco metrics, for consistency consideration, we re-run the code of Zhang 2020 on the benchmark data set instead of directly citing the numbers reported in Zhang 2020's paper.

---

### Official Review · Reviewer_4JHj · 2023-08-10

**Soundness:** 3

**Excitement:**

3: Ambivalent: It has merits (e.g., it reports state-of-the-art results, the idea is nice), but there are key weaknesses (e.g., it describes incremental work), and it can significantly benefit from another round of revision. However, I won't object to accepting it if my co-reviewers champion it.

**Paper Topic And Main Contributions:**

Paper Topic:
The paper focuses on extracting and structuring rich medical knowledge from multi-turn doctor-patient dialogues. This knowledge includes patient symptoms, doctor diagnoses, and prescribed medications. If effectively mined and represented, this medical knowledge can greatly benefit various clinical applications, such as diagnosis assistance and medication recommendations.


Main Contributions:
1. Extraction of Medical Information from Multi-turn Dialogues: One of the main contributions of the paper is addressing the task of "Dialogue Medical Information Extraction (DMIE)" from multi-turn doctor-patient dialogues. This task aims to detect predefined medical items (e.g., symptoms, surgeries) and their corresponding statuses (positive, negative) in the dialogues.

2. Heterogeneous Graph Modeling and Attention-based Modules: Unlike conventional approaches, the paper proposes the use of a heterogeneous graph to model relationships between medical items and statuses. Additionally, the paper introduces two consecutive attention-based modules that enrich the representation of medical items by incorporating both the dialogue context and status information. This approach allows for modeling the relationships among medical items and statuses in the DMIE task.

3. Performance Enhancement and State-of-the-Art Results: Experimental results demonstrate that the proposed model outperforms previous methods and achieves state-of-the-art performance. Particularly, the model shows superior performance on a public benchmark dataset compared to prior works.

**Questions For The Authors:**

Is there any previous work that uses a status-aware item refining module for the DMIE task? Is this the FIRST effort to deviate from formulating DMIE as multi-label classification problems?

Is there a specific reason for not including research on graph convolution in the related work section?

**Reasons To Accept:**

1. One of the notable strengths of the paper lies in the robustness of its experimental findings. The results obtained from the MIEACL benchmark dataset demonstrate a significant superiority of our model over existing state-of-the-art alternatives, thereby achieving an impressive new level of state-of-the-art performance. This outcome underlines the effectiveness and potential real-world applicability of the proposed model.

2. The paper is well-written and provides detailed explanations of the model, making it easy to follow and understand.

**Reasons To Reject:**

1. The authors assert the introduction of a novel model for the DMIE task, emphasizing interactions among medical items and the incorporation of doctor-patient dialogue information into item representations, along with an exploration of item-status matching relationships. However, I find the contributions to be relatively weak. While the use of a heterogeneous graph and attention-based modules holds potential, it lacks a strong selling point that sets it apart from similar approaches. Additionally, it's worth noting that this isn't the first instance of attempting the DMIE task. The paper could benefit from further clarifying how their approach distinguishes itself from existing methods and addressing the lack of novelty in certain aspects of the proposal.

2. Limited evaluation of recent graph methods: The paper does not include a comprehensive evaluation of recent graph methods, which may limit the understanding of the proposed model's performance relative to the state-of-the-art. Moreover, comparisons with only seven baselines can raise concerns.

3. Insufficient investigation of related research and inadequate referencing.

**Reproducibility:**

4: Could mostly reproduce the results, but there may be some variation because of sample variance or minor variations in their interpretation of the protocol or method.

**Reviewer Confidence:**

4: Quite sure. I tried to check the important points carefully. It's unlikely, though conceivable, that I missed something that should affect my ratings.

---

> ### Author Rebuttal · Authors · 2023-08-29
>
> We thank the reviewer 4JHj for the valuable comments. Please kindly find the point-to-point responses below.
>
> **Q1**: Is there any previous work that uses a status-aware item refining module for the DMIE task? Is this the FIRST effort to deviate from formulating DMIE as multi-label classification problems?
>
>
> **A1**: We have reviewed relevant papers on the DMIE task again and found few method that uses a status-aware item refining module to enhance doctor-patient dialogue representation. For those DMIE related papers, we found that the "A Speaker Aware Co Attention Framework for Medical Dialogue Information Extraction" also introduces an item representation module, however it does not fully model the internal connections among items and the interaction between items and status.
>
> The existing works mainly formulate DMIE as a classification task, using only dialogue information (speakers and utterances). While our proposed method explicitly models the relationship between dialogue, item, and status, and fully utilizes prior knowledge of the co-occurrence relationship between items, resulting in improved performance over existing works.
>
>
> **Q2**: Is there a specific reason for not including research on graph convolution in the related work section?
>
> **A2**: Thanks for your great advices. We will add more discussion on graph convolution models in the related work section. In this paper, we use the graph convolution model to mine the relations among items which in turn improves the performance of the DMIE task. The graph convolution model itself is not major focus of this paper.
>
> Following your great advices, in addition to the graph attention network [1] used in this paper, we conduct experiments on three recent graph works[2][3][4]. From the table below, we can find that equipping with any of these four graph models, the proposed method consistently outperforms SOTA works.
>
>
> |     **Model**     | **Macro Metric**| | | **Micro Metric**| | |
> | :----------- | :-----------: | :-----------: | :-----------: | :-----------: | :-----------: | :-----------: |
> |        | Precision| Recall | F1 | Precision| Recall | F1 |
> | CorNet       | 73.07 |71.43 |71.00| 67.72| 63.72| 65.66|
> | MIE        | 69.59| 65.89| 66.05| 69.43| 61.77| 65.38|
> | IIHGC with [1]        | 82.77 | 83.60 | 82.54 | 81.35 | 81.89 | 81.62 |
> | IIHGC with [2]        | 82.56 | 83.54 | 82.35 | 81.43| 81.40 | 81.41 |
> | IIHGC with [3]        | 82.42| 82.65 | 81.74 | 81.72| 81.24 | 81.48 |
> | IIHGC with [4]        | 81.15| 83.76 | 81.67 | 81.15| 81.69 | 81.41 |
>
>
> 1. Graph attention networks, ICLR 2018.
> 2. Principal Neighbourhood Aggregation for Graph Nets, NeurIPS 2020.
> 3. Simple and Deep Graph Convolutional Networks, ICML 2020
> 4. Training Graph Neural Networks with 1000 Layers, ICML 2021

---

### Meta-Review · Senior_Area_Chairs · 2023-10-05

**Recommendation:** 3

**Metareview:**

The reviewers appreciated the idea and the results shared in this paper but had mixed excitement scores. I believe that the suggested edits would be helpful to improve the current version of the paper. The authors are encouraged to:

- include the results added during the rebuttal and add a comprehensive analysis,

- better situate their contribution and correct the references as suggested by R1,

- explain how term variations are handled (see comment by R3 after the rebuttal),

- The limitations are the ethics sections should also be moved to the main text.

---

### Meta-Review · Area_Chair_rhqY · 2023-10-05

**Recommendation:** 3

**Metareview:**

This work proposed improved methods for the dialogue medical information extraction (DMIE) task, resulting in non-negligible improvements over the current best-performing systems on the corresponding dataset. Specifically, they develop a heterogenous graph modeling procedure to account for interactions between the different extraction types and attention models to augment the input representation to better include dialogue context and relationships between medical terms in the dialogue. Additional ablation studies are performed to elucidate additional relevant issues and add to understanding of the dynamics of the proposed method.

== Quality == The strongest element of the paper is the strength of the primary empirical results. The proposed method makes intuitive sense and is shown to perform well on the DMIE datasets, pushing the SotA forward.  Additionally, the paper is well-structured and easy to understand. Both reviewer 4JHj and DNoq suggest additional experiments that were provided during the rebuttal period. I would recommend including some of these results in the camera ready version (even if in an appendix if necessary) as they further support the efficacy of the proposed method.

== Clarity == Overall, the paper is well-motivated, well-structured, and easy to understand. While it would benefit from general polishing of the writing, it is well-written overall, but i recommend adding the additional results, references, and discussion as recommended by the reviewers.

== Originality == The authors claims methodological innovation, but it is really a case of applying more recent methodologies to this particular application. While this is still important (and they got it to work), they are not describing new methods that are easily adaptable to other problems, etc. This, it is a targeted innovation.

== Significance == Medical information extraction is an important and potentially impactful problem. From an application perspective, the proposed method achieves strong empirical results, pushing the SotA forward and thus will minimally be used as a baseline for future work. However, it is a domain-specific improvement using existing methods and doesn't make algorithmic advances that would be useful to the broader community.

---

### Decision · Program_Chairs · 2023-10-07

**Decision:**

Accept-Findings

**Comment:**

The reviewers appreciated the idea and the results shared in this paper but had mixed excitement scores. I believe that the suggested edits would be helpful to improve the current version of the paper. The authors are encouraged to:

- include the results added during the rebuttal and add a comprehensive analysis,

- better situate their contribution and correct the references as suggested by R1,

- explain how term variations are handled (see comment by R3 after the rebuttal),

- The limitations are the ethics sections should also be moved to the main text.|This work proposed improved methods for the dialogue medical information extraction (DMIE) task, resulting in non-negligible improvements over the current best-performing systems on the corresponding dataset. Specifically, they develop a heterogenous graph modeling procedure to account for interactions between the different extraction types and attention models to augment the input representation to better include dialogue context and relationships between medical terms in the dialogue. Additional ablation studies are performed to elucidate additional relevant issues and add to understanding of the dynamics of the proposed method.

== Quality == The strongest element of the paper is the strength of the primary empirical results. The proposed method makes intuitive sense and is shown to perform well on the DMIE datasets, pushing the SotA forward.  Additionally, the paper is well-structured and easy to understand. Both reviewer 4JHj and DNoq suggest additional experiments that were provided during the rebuttal period. I would recommend including some of these results in the camera ready version (even if in an appendix if necessary) as they further support the efficacy of the proposed method.

== Clarity == Overall, the paper is well-motivated, well-structured, and easy to understand. While it would benefit from general polishing of the writing, it is well-written overall, but i recommend adding the additional results, references, and discussion as recommended by the reviewers.

== Originality == The authors claims methodological innovation, but it is really a case of applying more recent methodologies to this particular application. While this is still important (and they got it to work), they are not describing new methods that are easily adaptable to other problems, etc. This, it is a targeted innovation.

== Significance == Medical information extraction is an important and potentially impactful problem. From an application perspective, the proposed method achieves strong empirical results, pushing the SotA forward and thus will minimally be used as a baseline for future work. However, it is a domain-specific improvement using existing methods and doesn't make algorithmic advances that would be useful to the broader community.